# Quality of Life Outcomes for Patients Who Underwent Conventional Resection and Liver Transplantation for Locally Advanced Hepatoblastoma

**DOI:** 10.3390/children10050890

**Published:** 2023-05-16

**Authors:** Zishaan Farooqui, Michael Johnston, Emily Schepers, Nathalie Brewer, Stephen Hartman, Todd Jenkins, Alexander Bondoc, Ahna Pai, James Geller, Gregory M. Tiao

**Affiliations:** 1Department of Surgery, University of Cincinnati College of Medicine, CARE/Crawley Building, Suite E-870 Eden Avenue, Cincinnati, OH 45267, USA; farooqzn@ucmail.uc.edu (Z.F.); johnstm3@ucmail.uc.edu (M.J.);; 2Division of Pediatric General and Thoracic Surgery, Cincinnati Children’s Hospital, 3333 Burnet Avenue, Cincinnati, OH 45229, USA; 3Behavioral Medicine & Clinical Psychology, Cincinnati Children’s Hospital, 3333 Burnet Avenue, Cincinnati, OH 45229, USA; 4Division of Oncology, Cincinnati Children’s Hospital Medical Center, Cincinnati, OH 45229, USA

**Keywords:** hepatoblastoma, orthotopic liver transplantation, quality-of-life

## Abstract

Hepatoblastoma is the most common malignant liver tumor of childhood, with liver transplant and extended resection used as surgical treatments for locally advanced tumors. Although each approach has well-described post-operative complications, quality-of-life outcomes have not been described following the two interventions. Long-term pediatric survivors of hepatoblastoma who underwent conventional liver resection or liver transplantation at a single institution from January 2000–December 2013 were recruited to complete quality-of-life surveys. Survey responses for the Pediatric Quality of Life Generic Core 4.0 (PedsQL, n = 30 patient and n = 31 parent surveys) and Pediatric Quality of Life Cancer Module 3.0 (PedsQL-Cancer, n = 29 patient and n = 31 parent surveys) were collected from patients and parents. The mean total patient-reported PedsQL score was 73.7, and the parent-reported score was 73.9. There were no significant differences in scores on the PedsQL between patients who underwent resection compared to those who underwent transplantation (*p* > 0.05 for all comparisons). On the PedsQL-Cancer module, procedural anxiety scores were significantly lower for patients who underwent resection as compared to transplant (M = 33.47 points less, CI [−60.41, −6.53], *p*-value 0.017). This cross-sectional study demonstrates that quality of life outcomes are overall similar among patients receiving transplants and resections. Patients who received a resection reported worse procedural anxiety.

## 1. Introduction

Hepatoblastoma (HB) is the most common primary liver malignancy in children [1,2,3,4], with increasing incidence in the United States [2,3,5]. It is most frequently diagnosed in children younger than 3 years of age [6]. The staging system for HB is based on pre-treatment extent of disease or PRETEXT staging. The goal of curative intent treatment is complete surgical resection [7]. Management of patients with locally advanced disease includes combination neoadjuvant chemotherapy and surgery.

Complete resection is critical for achieving curative treatment [7]; however, 50–60% of patients present with a disease considered unresectable at diagnosis [7,8]. Current management is based on the PRE-treatment EXTent of disease (PRETEXT) stage which classifies tumors into one of four groups based on the number of tumor-free liver sections. PRETEXT III and IV tumors are locally advanced and receive neoadjuvant chemotherapy. The response to chemotherapy is re-assessed according to the POST-treatment EXTent (POST TEXT) stage. Conventional resection with hepatectomy is recommended for patients whose tumor has down-staged to POSTTEXT I, II, and III without major vascular involvement. Extended treatment with chemotherapeutic regimens [8,9] beyond four cycles to allow for tumor resectability may induce drug resistance and exacerbate toxicities from chemotherapy [10,11]. After neoadjuvant chemotherapy, approximately 20% of tumors remain unresectable and without local control, these patients would succumb to the disease. The addition of liver transplantation as an oncologic option has dramatically altered the outcomes for these patients, improving their survival rate to >80% [12]. As a result, the frequency of liver transplantation for hepatoblastoma has increased significantly over the past two decades [13].

Both surgical treatment options (hepatectomy and liver transplantation) achieve similar remission rates and five-year overall survival [12]. Long-term sequelae differ significantly between resection and transplant. In addition to the adverse effects from chemotherapy, including ototoxicity and nephrotoxicity, children undergoing transplantation will require life-long immunosuppression, which carries with it unique comorbidities. [14,15]. The majority of patients received adjuvant chemotherapy. A transplant requires life-long immunosuppression, which independently carries a significant healthcare burden [1]. Improving survivorship for patients with HB brings new concerns regarding the long-term impact of the disease and treatment. Beyond medication effects, health-related quality of life (HRQOL) has emerged as an important measure in the overall outcome of patients with chronic diseases, including cancer.

In addition, children with cancer often have decreased quality of life (QoL). An important aspect of managing these patients is assessing their health-related quality of life. This involves understanding the emotional, social, and functional impact of their disease. The Pediatric Quality of Life Inventory 4.0 (PedsQL) and PedsQL Cancer Module are two frequently used assessment tools that have been validated for use in pediatric cancer and transplant patients [16,17].

The Pediatric Quality of Life Inventory 4.0 (PedsQL was developed to assess pediatric health-related QoL in a standardized fashion [17]. The score consists of 23 items applicable for healthy school and community populations, as well as pediatric populations with acute and chronic health conditions [16,17,18]. Furthermore, the PedsQL Cancer Module is tailored to QoL outcomes specifically for children with cancer [16,19,20,21,22,23,24,25,26]. These scores are now the most frequently used assessment tools in pediatric cancer and following pediatric liver transplants, respectively [18].

Patients with advanced and relapsed HB often require more intensive chemotherapy and have an increased risk of relapse. Those that undergo transplants will have significant lifetime healthcare concerns. The decision regarding treatment strategy is entirely governed by patient factors; therefore, quality-of-life outcomes do not dictate the choice of therapy. Each patient receives a resection or transplant based on tumor biology and resectability. However, the quality of life for children with advanced hepatoblastoma in both transplant and conventional resection groups has not been described in the literature. In this study, we evaluated health-related quality-of-life measures for patients with advanced hepatoblastoma that required conventional resection or liver transplantation. We hypothesized that there is no difference in health-related quality-of-life measures among treatment strategies.

## 2. Materials and Methods

### 2.1. Patient Population

From 2000–2013, patients with unresectable hepatoblastoma at initial diagnosis and any patient who underwent conventional resection via hepatectomy or liver transplantation for the primary diagnosis of hepatoblastoma were identified at CCHMC through existing oncology clinics and surgical databases. Parents of patients alive two years or more after completion of chemotherapy were approached for completion of QoL questionnaires. Exclusion criteria were patients receiving chemotherapy in the last 2 years, unwillingness or inability to participate, severe cognitive impairment rendering quality of life impossible to assess, and non-English speaking families. Study participation was voluntary. Prior to participation, participants signed written informed consent. Data were collected anonymously, and participants were informed of anonymity. The study was approved by the Institutional Review Board under study number # 2020-0333.

### 2.2. Survey Measures

Patients completed the age-appropriate version of the Peds QL 4.0 (Quality of Life) Generic Core Scales and the PedsQL Cancer module version 3.0. The Peds Quality of Life (QL) 4.0 Generic Core Scales is a 23-item assessment that incorporates four scales: (1) physical functioning (8 items), (2) emotional functioning (5 items), (3) social functioning, and (4) school functioning. This tool offers parallel child and parent-proxy report formats, which are designed to assess parents’ perception of their child’s HQROL. The responses are captured using Likert response scales that are slightly modified to be age appropriate. Higher PedsQL 4.0 scores indicate better QoL. The Peds QL Cancer Module is a 27-item survey assessment covering 8 scales: (1) pain and hurt (2 items), (2) nausea (5 items), (3) procedural anxiety (3 items), (4) treatment anxiety (3 items), (5) worry (3 items), (6) cognitive problems (5 items), (7) perceived physical appearance, (3 items), and (8) communication (3 items). The formatting, instructions, Likert response scale, and scoring are the same for both surveys. Similar to PedsQL Generic Score Scales, higher scores are indicative of better HRQOL.

Procedures: questionnaires were mailed to all participants in June 2016 with up to three reminders mailed to all non-responders. Study data were collected up until November 2022 and managed using REDCap electronic data capture tools hosted at Cincinnati Children’s Medical Center [27]. The answers from the questionnaires were scored as per the respective protocol as established and described by the questionnaire developers.

Statistical analysis: categorical variables were described as frequencies and continuous variables as medians with interquartile ranges (IQR), or as means with standard deviations (SD). We used a *t*-test for independent samples to compare continuous variables (i.e., quality of life scores) across treatment groups. Multiple linear regression was modeled with subscale score as a continuous outcome controlled for surgery type (conventional resection vs. transplant), age, and gender. Data were analyzed using JMP^®^ Pro Version 16.0.0 software as well as SAS.

## 3. Results

A total of 70 patients were identified; 29 were deceased or lost to follow-up. Thirty-four agreed to participate (82.9% out of the 41 remaining). Figure 1 demonstrates a flow chart of participants. Of the 34 recruited patients, 31 parents (91%) completed both PedsQL 4.0 Generic Core Scales and PedsQL Cancer questionnaires. A total of 17 patients had orthotopic liver transplants (OLT) and 14 had hepatic resections. A total of 30 patients (88%) completed the PedsQL 4.0 Generic Core Scales, and 29 patients (85%) completed the PedsQL Cancer survey. Data were retained for one patient who did not complete PedsQL Cancer because these measures were ultimately analyzed separately. Sensitivity analysis done whilst excluding that patient did not yield a significant difference in results. A total of 81% of all respondents (i.e., those who completed any survey) were between the ages of 8 to 18 years. As shown in Table 1, a total of 52% of participants were male, and 48% were female. 17 (55%) underwent transplants and 14 (45%) underwent conventional resection. The majority of patients were white (n = 29, 93.5% of those who responded with race). There was no significant difference in the distribution of PRETEXT stage between patients undergoing resection or transplant (*p* = 0.12), nor was there a difference in whether neoadjuvant therapy was administered (*p* = 0.77). Recurrence was present in three patients; two were in the resection group and one in the transplant group.

### 3.1. Peds QL 4.0 Generic Core Scales

We first analyzed patient quality of life on a broad, generic scale in all patients, as well as parents’ assessment of their child’s quality of life following surgery. As shown in Table 2, the mean total score on patient-reported surveys was 73.7, and for parent-reported surveys, the mean was 73.9. There were no significant differences between parent and child self-reporting on the surveys in either the total score or the individual domains of physical functioning, emotional functioning, social functioning, or school functioning (*p* > 0.30 for all comparisons). The largest impact on functioning overall was seen at school, with the lowest survey scores being related to function at school. The patient-reported school functioning score was 63.8, while the parent-reported school functioning score was 66.3. As shown in Table 3, there were no significant differences in functioning scores between patients who underwent conventional resection for hepatoblastoma compared to those who underwent transplantation (*p* > 0.05 for all comparisons).

### 3.2. Peds QL 3.0 Cancer Module

Sixteen patients who underwent liver transplantation and 13 patients who underwent resection completed the PedsQL 3.0 Cancer. A total of 17 parents of transplant recipients and 14 parents of children who underwent conventional resection also completed this survey. As shown in Table 4, the domains of pain, nausea, worry, and perceived physical appearance all had scores >70. There were no significant differences between the transplant group and the conventional resection group. Overall, patients who underwent transplants reported lower scores for procedural and treatment anxiety, although most scores did not show a statistically significant difference. The difference was most marked in procedural anxiety, where there was a significant difference between patients who underwent transplants (mean 65.6) vs. conventional resections (mean 27.6, *p* < 0.01). This significant difference was maintained when analyzing parent-reported surveys as well (*p* = 0.03).

As scores between patient and parent were highly correlated, a multiple linear regression model was performed on the subset of patients only, correcting for gender and age. This was performed on both the Peds QL 4.0 Generic Core Scales as well as the Peds QL 3.0 Cancer Module, as shown in Table 5. Survey scores for procedural anxiety were significantly lower for conventional resection patients as compared to transplant patients; on average, a patient who underwent conventional resection scored −33.47 points less on the procedural anxiety subscale in the Peds QL 3.0 Cancer module (CI [−60.41, −6.53], *p*-value 0.017).

## 4. Discussion

Long-term survival is now the rule rather than the exception following pediatric liver transplantation [23]. One-third of ten-year survivors of pediatric liver transplant in the Pediatric Liver Transplantation registry did not report any immunosuppression-induced complications, including reduced growth velocity, post-transplant lymphoproliferative disease (PTLD), hypercholesterolemia, or renal dysfunction [23]. Quality-of-life analyses have not been reported in patients who have undergone treatment for hepatoblastoma. Given the viability of both conventional resection and transplant as options for hepatoblastomas that were unresectable at diagnosis, this study aimed to assess patients’ quality of life following these two surgical strategies.

There are few publications reporting HRQoL for pediatric cancer survivors. In an Australian study, they had 182 parents and children complete QoL surveys. Overall, there was no difference compared to population norms, however, a subset of patients have a persistent significant decrease in HRQOL [24]. A Swiss study found worse reported QoL in survivors of CNS tumors, neuroblastoma, and relapse [25]. Prior work studying the QoL in pediatric liver transplant survivors has demonstrated that there is an overall negative effect on QoL and that this varies widely [26,27,28]. No prior studies were found evaluating liver transplantation and hepatic resection in pediatric patients with hepatoblastoma. This is the first study to evaluate quality of life in children with hepatoblastoma, and this uniqueness is further bolstered by the use of the validated measures of Peds QL Generic Core and Peds QL Cancer 3.0.

### 4.1. Practical Clinical Implications

In our population, the mean patient-reported total score in the Peds QL Generic Core questionnaire was 73.7, which is comparable to previous reports by Varni et al. in a pediatric oncology population [16]. The authors compared the total scores and subdomain scores of oncology patients (mean total score = 72.2) to age-matched healthy patients (mean total score = 83.0) and found that, as expected, overall quality-of-life measures were worse in the oncology group. As demonstrated in Figure 2A, both our transplant and conventional resection patients had PedsQL Generic scores that were comparable to the previously published oncology cohort [16]. Consistent with other published work [16,18,29], school functioning scores tend to be low (mean score = 63.8 in our patient population, 68.5 in the oncology cohort, and 79.4 in normal controls). Importantly, given the small sample size, the present study does not compare to age-matched healthy patients; however, school functioning scores are consistently low across the literature, and this strongly suggests that falling behind in school is a pain point in quality of life following treatment in hepatoblastoma patients treated by either conventional resection or transplant. Figure 2B shows the Peds QL Cancer 3.0 scores in our cohort in relation to the same oncology cohort by Varni et al. Although our study did not perform a statistical comparison with the prior Varni cohort, we still see large differences in scores in our patient population in the sub-scores of cognitive problems, treatment anxiety, and procedural anxiety.

Table 3, Table 4 and Table 5 demonstrated that transplant and conventional resection patients reported similar scores in most of the quality-of-life outcomes; however, patients who underwent conventional resection had worse scores on a clinically relevant quality-of-life measure: anxiety associated with procedures. This was borne out in both patient-reported surveys as well as parent-reported surveys. This was a surprising finding given the common perception that transplantation negatively impacts the long-term quality of life [1].

Our dataset did not evaluate the differences in office visits, re-operations, and other procedures between patients who underwent transplantation and conventional resection. We speculate that transplant patients in the immediate and short-term post-operative period undergo a higher number of procedures and become habituated to procedures. The overall finding of increased anxiety with procedures is a potentially clinically impactful finding worthy of future investigation. The involvement of child life specialists or psychologists in the care of patients receiving resections could provide strategies to effectively manage procedural distress.

It should be emphasized that although we have compared quality-of-life outcomes among hepatoblastoma patients undergoing resection and transplantation, this is not meant to be interpreted as a suggestion of the superiority of one treatment over another. Patients receive liver transplantation because they are deemed clinically unresectable. The question of quality of life is secondary at the time of treatment decisions. Therefore, our findings support the hypothesis that there is no overall difference in quality-of-life measures with the exception of anxiety associated with procedures. However, this cannot be used to guide treatment decisions.

### 4.2. Limitations

This study has important limitations. Despite its increasing incidence, hepatoblastoma remains a rare tumor [14,30]. Survey response rates of this retrospective single-center study were somewhat lower than in comparable publications [31], limiting the capacity for statistical analysis. The low response rate may be owed to multiple factors, including data gathering through online media (REDCap) as well as multiple questionnaires. Both the PedsQL and PedsQL Cancer Module consist of 23 questions each. Likewise, both questionnaires comprise parent as well as patient portions. Only 50% of patients completed all four questionnaires. The most important consequence of this study is the lack of significant findings in subdomains on quality-of-life scores. Nonetheless, the current study provides evidence of clinical equipoise between conventional resection and transplant, indicating that transplantation may not be worse for quality of life in patients with hepatoblastoma. It is possible that a higher-powered study may find more significant differences in addition to the difference in procedural anxiety found in the present analysis.

Another limitation is the variability in age from the time of treatment. This is certainly a limitation, but there was no difference in age among resection and transplant patients in the bivariate analysis and there was no effect of age on QoL in a sensitivity analysis but this is likely underpowered. Additionally, the risk of hepatoblastoma is increased in patients with syndromic disease including Beckwith-Wiedemann Syndrome and familial adenomatous polyposis coli. Likewise, low birthweight has been consistently associated with an increased risk of hepatoblastoma. While any of these may conceivably impact the long-term quality of life, the present study did not control for age and comorbidities in order to avoid multiple-testing bias in a small sample. For the same reasons, we did not correct for chemotherapy regime.

## 5. Conclusions

In summary, the treatment of hepatoblastoma is rapidly evolving, with the addition of transplantation as a viable strategy for tumors that are unresectable after neoadjuvant chemotherapy. The present cross-sectional study demonstrates that quality-of-life outcomes in hepatoblastoma patients were similar among treatment strategies. In one domain—anxiety associated with procedures—scores were improved in post-transplant patients compared to those who underwent conventional resection. The overall quality of life scores was comparable in hepatoblastoma patients compared to other children who survive cancer. Larger multi-center studies are needed to confirm this observation and to better characterize the long-term quality-of-life outcomes in patients with hepatoblastoma.

## Figures and Tables

**Figure 1 children-10-00890-f001:**
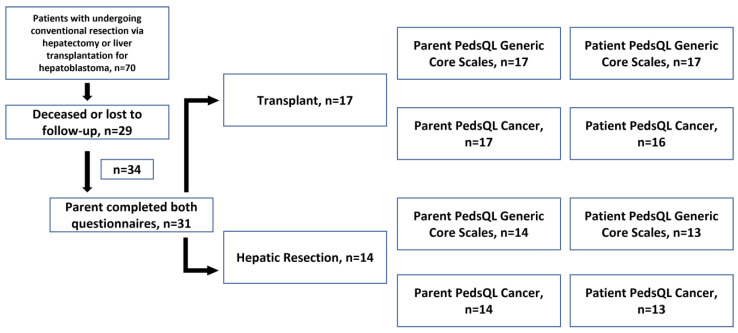
Participant flowchart.

**Figure 2 children-10-00890-f002:**
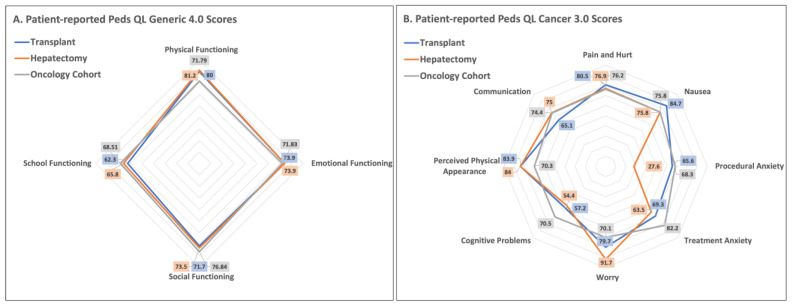
Radar plot of patient-reported scores (**A**) Patient-reported Peds QL Generic 4.0 Scores: Gray line is scores from a previously published oncology cohort [16]; (**B**) Patient-reported Peds QL Cancer 3.0 Scores: Blue, orange, and gray lines with similar meanings as (**A**). *Of note, the data for Oncology Cohort was reprinted/adapted with permission from Ref.* [16] *2004, Varni et al.*.

**Table 1 children-10-00890-t001:** Demographic characteristics.

	All Patients, n = 31	Transplant, n = 17	Resection, n = 14	*p*-Value
Operation performed, n (%)				
Transplant	17 (54.8%)	-	-	
Resection	14 (45.2%)	-	-	
Age (years) at time of surgery, median (IQR)	1.9 (1.4–2.7)	1.7 (0.97–2.0)	2.3 (1.6–3.0)	0.08
Age at time of survey by categories. n (%)				
5 to 7 years	5 (16.1%)	1 (5.9%)	4 (28.6%)	0.06
8 to 12 years	14 (45.2%)	6 (35.3%)	8 (57.1%)	
13 to 18 years	11 (35.5%)	9 (52.9%)	2 (14.3%)	
18 to 25 years	1 (3.2%)	1 (5.9%)	0	
Sex, n (%)				
Male	16 (51.6%)	10 (52.6%)	8 (57.1%)	0.38
Female	15 (48.4%)	7 (36.8%)	6 (42.9%)	
Ethnicity				
White	29 (93.5%)	17 (100%)	12 (85.7%)	0.27
Black	1 (3.2%)	0	1 (7.1%)	
Missing	1 (3.2%)	0	1 (7.1%)	
PRETEXT				
I	1 (3.2%)	1 (5.9%)	0 (0%)	0.12
II	11 (35.5%)	3 (17.7%)	8 (57.1%)	
III	10 (32.3%)	7 (41.2%)	3 (21.4%)	
IV	1 (3.2%)	1 (5.9%)	0 (0%)	
Missing	8 (25.8%)	5 (29.4%)	3 (21.4%)	
Neoadjuvant chemotherapy given				
Yes	19 (61.3%)	9 (52.9%)	10 (71.4%)	0.77
No	5 (16.1%)	2 (11.8%)	3 (21.4%)	
Missing	7 (22.6%)	6 (35.3%)	1 (7.1%)	

Pearson Chi-square test for categorical variables. Median test for continuous variables.

**Table 2 children-10-00890-t002:** Mean scores of PedsQL 4.0 Generic Core Scales.

	Patient Survey (n = 30)	Parent Survey (n = 30)	
	Mean (SD)		*p*-value
Total Scale Score	73.7 (18.8)	73.9 (18.1)	0.89
Subscale			
Physical Functioning	80.5 (23.4)	79.9 (20.9)	0.50
Emotional Functioning	73.9 (22.1)	71.0 (22.7)	0.14
Social Functioning	72.5 (23.9)	74.6 (23.4)	0.38
School Functioning	63.8 (20.3)	66.3 (19.2)	0.46
Psychosocial Summary Score	70.1 (18.7)	70.6 (19.3)	0.60

**Table 3 children-10-00890-t003:** Differences in PedsQL 4.0 Generic Core Scales as a function of surgery.

	Patient Responses			Parent Responses		
	Transplant n = 17	Resection n = 13		Transplant n = 17	Resection n = 14	
	Mean (SD)		*p*-value	Mean (SD)		*p*-value
Total Scale Score	73.0 (18.2)	74.6 (20.2)	0.83	72.4 (18.9)	75.6 (17.6)	0.63
Subscale						
Physical Functioning	80.0 (20.7)	81.2 (27.3)	0.89	78.6 (17.8)	81.5 (24.8)	0.72
Emotional Functioning	73.9 (22.7)	73.9 (22.0)	0.99	70.6 (23.0)	71.4 (23.2)	0.92
Social Functioning	71.7 (23.1)	73.5 (25.8)	0.84	70.3 (24.5)	79.7 (21.9)	0.27
School Functioning	62.3 (20.0)	65.8 (21.4)	0.65	66.7 (23.7)	65.7 (12.5)	0.88
Psychosocial Summary Score	69.3 (18.7)	71.1 (19.4)	0.8	69.2 (21.0)	72.3 (17.6)	0.66

**Table 4 children-10-00890-t004:** Differences in PedsQL Cancer 3.0 scores as a function of surgery.

	Patient Responses			Parent Responses		
	Transplant, n = 16	Resection, n = 13		Transplant, n = 17	Resection, n = 14	
	Mean (SD)		*p*-value	Mean (SD)		*p*-value
Total	73.1 (16.1)	68.0 (17.3)	0.41	69.1 (19.5)	69.5 (13.9)	0.94
Dimension Scores						
Pain and Hurt	80.5 (21.4)	76.9 (25.4)	0.69	79.4 (24.2)	75 (24.5)	0.62
Nausea	84.7 (16.7)	75.8 (24.2)	0.25	84.4 (19.9)	84.3 (18.0)	0.99
Procedural Anxiety	65.6 (30.6)	27.6 (34.1)	0.004 *	60.3 (29.8)	33.3 (37.3)	0.03 *
Treatment Anxiety	69.3 (24.3)	63.5 (31.6)	0.58	71.6 (28.6)	50 (32.7)	0.06
Worry	79.7 (20.4)	91.7 (10.2)	0.06	76.5 (25.0)	88.1 (13.8)	0.13
Cognitive Problems	57.2 (26.7)	54.4 (27.4)	0.79	48.8 (29.6)	60.7 (19.8)	0.21
Perceived Physical Appearance	83.9 (20.7)	84.0 (20.0)	0.99	75.0 (24.3)	83.3 (19.9)	0.31
Communication	65.1 (26.7)	75 (30.6)	0.36	62.7 (29.2)	79.2 (30.4)	0.14

* denotes significance, *p*-value < 0.05.

**Table 5 children-10-00890-t005:** Multiple linear regression for PedsQL Patient General and Cancer Scores.

	𝛽_surgery_ *	SE	95% Confidence Interval	*p*-Value
PedsQL General Survey				
Total Scale Score	3.70	7.66	(−12.19, 19.60)	0.63
Subscale				
Physical Functioning	0.11	9.50	(−19.60, 19.82)	0.99
Emotional Functioning	4.73	9.51	(−15.00, 24.45)	0.62
Social Functioning	2.69	9.73	(−17.49, 22.88)	0.78
School Functioning	9.60	8.70	(−8.44, 27.63)	0.28
Peds QL Cancer 3.0 Dimension				
Pain and Hurt	−2.79	8.50	(−20.48, 14.89)	0.75
Nausea	−6.91	8.33	(−24.24, 10.41)	0.42
Procedural Anxiety	−33.47	13.00	(−60.41, −6.53)	0.017 *
Treatment Anxiety	−3.86	11.44	(−27.64, 19.93)	0.74
Worry	10.11	7.31	(−5.09, 25.31)	0.18
Cognitive Problems	5.31	11.76	(−19.16, 29.77)	0.66
Perceived Physical Appearance	−3.50	9.28	(−22.80, 15.80)	0.71
Communication	4.92	13.52	(−23.19, 33.03)	0.72

𝛽 surgery * is parameter estimate for change in score. Reference surgery type is Transplant. Multiple regression corrected for gender and age.* denotes significance, *p*-value < 0.05.

## Data Availability

The data presented in this study are available on request from the corresponding author. The data are not publicly available due to patient privacy and ethical concerns.

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
