# Peer review of "Quality of Life Outcomes for Patients Who Underwent Conventional Resection and Liver Transplantation for Locally Advanced Hepatoblastoma"

_children, 2023, doi:10.3390/children10050890_

Round 1
Reviewer 1 Report
Dear Authors,
Congratulations on your extensive work, concerning Quality of Life Outcomes for Patients who Underwent Conven-2 tional Resection and Liver Transplantation for Locally Ad-3 vanced Hepatoblastoma
I suggest some major revisions:
Introduction
Lines 74-75: We sought to evaluate quality-of-life outcomes for long term survivors who under-74 went transplantation and surgical resection in patients who presented with PRETEXT III 75 or IV hepatoblastoma. Understanding quality of life outcomes is an important component 76 of a comprehensive discussion for families who face this challenging combination of ther-77 apeutic options.
I think the authors should formulate study aims with more details. Furthermore, some study hypotheses should be added at the end of this section.
M&M:
How about ethical issues.? Did authors obtain written consent forms from parents, and permission from a local Bioethic Comission? Were participants assured of anonymity?
How about adding a participant flowchart, showing recrutiment to the study in details??
Detailed data on Peds QL 4.0 88 (Quality of Life) Generic Core Scales and the PedsQL Cancer module version 3.0. is missing.
Did authors collect any clinical data regading treatment???
In general, M&M is too concise, lacks of many details.
Results;
Since the final study sample is n = 31, in my opinion the study title should be reformulated to: Quality of Life Outcomes for Patients who Underwent Conventional Resection and Liver Transplantation for Locally Advanced Hepatoblastoma: prelimainary report
As I said, some clinical description of patient is missing.
In addition, correlational analysis between clinical data and patients’ Quality of Life should be done.
Discussion:
The last part of discussion should be dvided into subsections: strenghts and limitations, future research implications, practical clinical implications, and then study conclusions.
In addition, verification of study hypotheses should be discussed.
Reviewer 2 Report
This study reports on the Quality of Life of pediatric patients undergoing either resection or liver transplantation for hepatoblastoma. I have a few remarks.
Methods: The authors report on a historic cohort of children, that can be twenty and more years older today compared to initial procedure. How reliable are the parameters (e.g. procedural anxiety, treatment anxiety) when the initial event was two decades ago and most likely, many children could not comprehend what was happening?
It might be helpful to briefly explain the metrics/scaling of the scores mentioned (PedsQL) to help the reader interpret the results.
Results:
It might be interesting, how much time passed since treatment and if recurrence was present in any case?
The similarity between patients´and parents´scores are striking. Is it not likely, that they interacted while filling out the questionnaires (e.g. children asking their parents: How was it back then?) Is there a way to differentiate?
In univariate analysis, only procedural anxiety and worry showed significant difference. What is the purpose of a multivariate analysis including parameters that did not show statistical significance in univariate analysis?
After adjusting for gender and sex, how many patients were left for inclusion for multivariate analysis?
Tx-patients tend to undergo a variety of re-interventions, hospital visits etc. during their course and all cancer patients undergo frequent aftercare. Is it possible to assess the impact of these ongoing medical requirements on individual patients?
Did the authors look for differences in a time-dependent manner?
Discussion:
From my opinion, a comparison for Quality of Life makes the most sense, when two e.g. treatment options are available. But, as the authors wrote in the beginning, patient received liver transplantation because they were deemed unresectable, meaning without resection, they would likely have passed. This should be discussed at least.
Conclusion: Again, I do not think, the different therapeutic regimens can be termed “treatment strategies”, as the Tx group was never feasible for another treatment option, leaving the treating physicians no choice, and thus, the question of quality of life was probably secondary at the time of treatment.
Round 2
Reviewer 1 Report
The authors have addressed all my remarks and concerns.
Reviewer 2 Report
I have no further comments, the authors have made a thorough revision.